# Post-COVID-19 Cognitive Decline and Apoe Polymorphism: Towards a Possible Link?

**DOI:** 10.3390/brainsci13121611

**Published:** 2023-11-21

**Authors:** José Wagner Leonel Tavares-Júnior, Danilo Nunes Oliveira, Jean Breno Silveira da Silva, Werbety Lucas Queiroz Feitosa, Artur Victor Menezes Sousa, Samuel Cavalcante Marinho, Letícia Chaves Vieira Cunha, Safira de Brito Gaspar, Carmem Meyve Pereira Gomes, Laís Lacerda Brasil de Oliveira, Caroline Aquino Moreira-Nunes, Emmanuelle Silva Tavares Sobreira, Maria Elisabete Amaral de Moraes, Manoel Alves Sobreira-Neto, Raquel Carvalho Montenegro, Pedro Braga-Neto

**Affiliations:** 1Neurology Division, Clinical Medicine Department, Faculty of Medicine, Federal University of Ceará (UFC), Fortaleza 60020-181, CE, Brazil; wagnerleoneljr@gmail.com (J.W.L.T.-J.); danilonunesoliveira@gmail.com (D.N.O.); werbetylucas@gmail.com (W.L.Q.F.); arturvmenezes@gmail.com (A.V.M.S.); leticiachavescunha@gmail.com (L.C.V.C.); emmanuelle_silvatavares@yahoo.com.br (E.S.T.S.); manoelsobreira@yahoo.com.br (M.A.S.-N.); 2Clinical Research Unit, Walter Cantidio University Hospital, Federal University of Ceará (UFC), Fortaleza 60020-181, CE, Brazil; 3Medicine Research and Development Center (NPDM), Pharmacogenetics Laboratory, Federal University of Ceará (UFC), Fortaleza 60020-181, CE, Brazil; jean-brenosilveira@hotmail.com (J.B.S.d.S.); laisbrasil@ufc.br (L.L.B.d.O.); carolfam@gmail.com (C.A.M.-N.); betemora@ufc.br (M.E.A.d.M.); rcm.montenegro@gmail.com (R.C.M.); 4Health Sciences Center, State University of Ceará (UECE), Fortaleza 60714-903, CE, Brazil; samuelcm98@hotmail.com (S.C.M.); safiradebritogaspar@gmail.com (S.d.B.G.); carmem.gomes@aluno.uece.br (C.M.P.G.)

**Keywords:** COVID-19, cognitive impairment, long-COVID, dementia, *APOE*

## Abstract

*APOE* ε4 polymorphism has been recently described as a possible association with cognitive deficits in COVID-19 patients. This research aimed to establish the correlation between COVID-19 and cognitive impairment, and the *APOE* gene polymorphism among outpatients. We performed a cross-sectional study with confirmed COVID-19 patients and neurological symptoms that persisted for more than three months from onset. *APOE* genotypes were determined. The final number of patients included in this study was 219, of which 186 blood samples were collected for *APOE* genotyping, evaluated 4.5 months after COVID-19. Among the participants, 143 patients (65.3%) reported memory impairment symptoms as their primary concern. However, this complaint was objectively verified through screening tests (Addenbrooke Cognitive Examination-Revised and Mini-Mental State Examination) in only 36 patients (16.4%). The group experiencing cognitive decline exhibited a higher prevalence of the *APOE* ε4 allele than the normal group (30.8% vs. 16.4%, respectively, *p* = 0.038). Furthermore, *the APOE* ε4 allele and anxiety symptoms remained significant after multivariate analysis. This study assessed an outpatient population where cognitive changes were the primary complaint, even in mild cases. Moreover, the ε4 allele, sleep disorders, and anxiety symptoms were more frequent in the cognitive decline group.

## 1. Introduction

COVID-19 exhibits a diverse range of clinical manifestations, including general neurological symptoms [1]. Furthermore, cognitive impairment can occur after COVID-19, either in the acute or chronic phases, regardless of COVID-19 clinical severity [2,3]. Subsequently, cognitive manifestations after the acute and subacute phases of the disease began to be reported even in patients with mild or asymptomatic forms of the disease. Such manifestations can generally occur with other symptoms, such as fatigue and sleep disorders, in a condition that has been called Long-COVID-19 [4]. In this sense, Brutto et al. evaluated outpatients with mild cases of the disease six months after infection and, using the MoCA, showed a decline in 21% of patients compared to data from the same patients before the pandemic [3].

Previous publications have suggested a possible role of *APOE* in conferring protection or risk of more severe clinical manifestations of COVID-19, with similar physiopathology processes already described in Alzheimer’s disease (AD) [5,6,7]. Kuo et al. linked more severe COVID-19 in subjects with the ε4 allele of the *APOE* gene. The authors of this study hypothesized that this finding might be related to the high level of expression of *APOE* genes together with angiotensin-converting enzyme 2 (ACE-2) in the alveolar cells of the lungs [6]. Another group of researchers studied 249 volunteers with an average age of 49 years and evidenced the E2 allele’s protective role against more severe COVID-19 clinical conditions [5]. Similarly, Zhang et al. evaluated 142 patients with COVID-19 and found that those with *APOE* E4 had elevated inflammatory factors [8]. In another study, Zorkina et al. did not find any influence of the baseline serological status for COVID-19 and the *APOE* gene polymorphism on cognitive rehabilitation in a sample of individuals over 65 years old measured through changes in Mini-Mental State Examination (MMSE) scores [9]. This association is significant, as the same allele confers a higher risk of sporadic Alzheimer’s disease (AD) [10]. An article published with the preliminary results of our study did not reveal an association between cognitive impairment and *APOE* [11].

Thus, the possible post-COVID-19 cognitive decline and the relationship between the *APOE* polymorphism and post-COVID-19 severe and cognitive conditions raise concerns regarding the subsequent development of neurodegenerative diseases, such as Alzheimer’s disease [12]. This research aimed to ascertain the link between Long-COVID-19-related cognitive impairment and *APOE* gene polymorphism in a larger sample of outpatients in a public university hospital IN Northeast Brazil.

## 2. Methods

### 2.1. Subjects

We conducted a cross-sectional study at an outpatient clinic for COVID-19 patients at Walter Cantídio University Hospital, Fortaleza, Northeast Brazil. Patients were recruited in July and August 2020 as part of our research team’s continuous prospective longitudinal study. We included patients with a single, confirmed COVID-19 diagnosis, by RT-PCR nasal swab or serological test, in the last twelve months before the research was carried out with any neurological symptom persisting for over three months since the onset. Two independent neurologists (JWLTJ and DNO) performed patient clinical evaluations. We did not evaluate the inter-examiner error. Identical clinical assessment and identification forms were administered to all participants. Various factors were assessed, including age, gender, educational background, initial neurological symptoms, hospitalization history, type of COVID-19 test administered, results of additional tests, current medical conditions, and details regarding alcohol and tobacco use. We did not determine the ethnic group of the patients. We also looked for control patients without COVID-19 infection, but unfortunately, the country was undergoing a severe health crisis during the pandemic, and patients without COVID-19 were afraid to participate in the research in a hospital environment.

### 2.2. Clinical and Cognitive Evaluation

Dyspnea levels were evaluated before and after the onset of COVID-19 using the Medical Research Council’s (MRC) dyspnea scale. Cognitive evaluations utilized standardized tools such as Addenbrooke’s Cognitive Examination-Revised (ACE-R), the Mini-Mental State Examination (MMSE), and the Clinical Dementia Rating (CDR). Functional abilities were evaluated using the Pfeffer instrumental activities of daily living scale. At the same time, mood was assessed through the Geriatric Depression Scale (GDS) or the Beck Inventory, depending on the patient’s age. The detailed methodology and specific cutoff points applied are outlined in the Appendix A. Mild cognitive impairment (MCI) was diagnosed in cases where cognitive complaints were confirmed through screening tests, even without associated functional impairment. Patients reporting cognitive concerns without objective impairment in the administered tests were categorized as experiencing subjective cognitive decline (SCD) [13]. We also grouped patients with DCS, MCI, and dementia under the general term cognitive decline (CD) to compare patients without cognitive complaints, which we called normal. Grouping patients with CD and the increase in the number of patients makes our current work different from our previous data. In our previous study, we only compared patients with MCI and dementia with patients with SCD and normal. We also grouped MCI and dementia under the term cognitive impairment (CI) to compare with patients without cognitive impairment. Scale cutoff scores applied are discriminated against in the Appendix A.

### 2.3. APOE Genotyping

The genotypes of *APOE* were identified through the application of real-time Polymerase Chain Reaction (qPCR). DNA sample quality was evaluated by nanodrop and Qubit2.0. *APOE* genotypes were determined by real-time Polymerase Chain Reaction (qPCR) using the TaqMan^®^ allelic discrimination system (TaqMan^®^ SNP Genotyping Assay, ThermoFisher^®^, Waltham, MA, USA) [14]. To this end, we used probes per the sequences provided by the manufacturer: C___3084793_20 (rs429358) and C____904973_10 (rs7412), observing the information contained in the catalog number 4351379 and similar protocols described in the literature for performing the technique. All samples were used. If the DNA sample was not pure, we re-extracted it from fresh blood samples collected in the following clinical appointment. The Appendix A also includes more technical details of the *APOE* survey. 

### 2.4. Statistical Analysis

Categorical data were expressed as absolute counts and percentages. The chi-square test was used to evaluate the association among categorical data. Continuous data were first evaluated for normal distribution using the Kolmogorov-Smirnov test [15]. Normal data were expressed as mean ± standard deviation, and non-normal data as median and interquartile range. Normal data were compared using a one-way ANOVA with Tukey’s post-test and a Kruskal-Wallis test with Dunn’s post-test for non-normal data [16]. Data were analyzed using SPSS software for Macintosh, version 23 (Armonk, NY, USA: IBM Corp.). Values of *p* < 0.05 were considered statistically significant. 

Moreover, we performed logistic regression analyses using DC as the dependent event. For the multivariate models, variables that presented *p* < 0.100 in the bivariate analysis were selected, along with possible confounders based on scientific criteria. The selected variables were exposed to the backward stepwise method. In short, all selected variables are included simultaneously in an initial model. Then, one by one, variables are removed using the highest *p*-value as a criterion in each model generated until a final model is reached with only variables presenting *p* < 0.20. Analyses were performed using SPSS software for Macintosh (Version 23.0; Armonk, NY, USA: IBM Corp.).

## 3. Results

Two hundred forty-one individuals were screened, of which 22 were disqualified (10 for lack of neurological symptoms, 10 for testing negative for COVID-19 in the tests, and two for being unable to submit to the application of the batteries) (Figure 1). Two hundred nineteen patients were finally included in the study, of which 186 provided blood samples for *APOE* genotyping, and all the following analysis was conducted. The evaluation of patients occurred approximately 4.5 months after their COVID-19 diagnosis. 

Table 1 provides a descriptive overview of the patients’ attributes. Women prevailed (64.8%), the mean age was 46.4 years (SD = 14.5), and most had more than eight schooling years (80.4%). Most patients (74.9%) were not hospitalized during the acute phase of the disease, and only a small percentage had a severe clinical condition requiring ICU admission (5.4%). The main complaint reported by one hundred forty-three patients (65.3%) was memory impairment. Nevertheless, this concern was validated through objective screening tests in 36 patients (16.4%). We identified new cases of dementia or the deterioration of existing dementia in 4.9% of the total sample among patients with cognitive impairment, with a mean age of 69.8 years observed in these patients. Thirty-eight patients (17.1%) had depression, six were diagnosed using the GDS, 32 using the Beck inventory, and 57 (25.7%) had persistent anxiety symptoms.

Table 2 compares sociodemographic, clinical, and post-COVID-19 symptom characteristics between the groups with dementia, MCI, DCS, and normal. The dementia group had a higher mean age than the others (69.8 years; *p* < 0.001). Table 3 compares sociodemographic and clinical characteristics and post-COVID-19 symptoms between groups with cognitive decline (CD) and normal. There was no difference between the groups regarding depression. The cognitive decline group had a higher frequency of anxiety symptoms than the normal group (30.8 vs. 17.1%, respectively, *p* = 0.028). The CD group also showed a higher frequency of sleep disorders than the normal group (35.7 vs. 17.1%, respectively, *p* = 0.004). There was no significant difference in the patients’ cognitive status regarding schooling or hospitalization. The cognitive decline group was older than the normal group (48 vs. 43 years, *p* < 0.001).

Table 4 reveals that the most prevalent *APOE* genotype was ε3/ε3, accounting for 65.9% of cases, with the ε3 allele predominating (96.7%). In the second place, the ε3/ε4 genotype represented 23.2% of all cases, while the ε4 allele was found in 25.9% of instances. The group experiencing cognitive decline exhibited a higher frequency of the *APOE* ε4 allele than the normal group (30.8% vs. 16.4%, respectively, *p* = 0.038) (Table 5). Additionally, the presence of the ε4 allele emerged as an independent risk factor for cognitive decline, with an odds ratio of 2.33 (Table 6). Furthermore, anxiety symptoms remained statistically significant even after multivariate analysis, with an odds ratio of 3.75 (Table 6). The MMSE and ACE-R scores according to the patients’ age group are shown in (Table 7). Regarding the comparison of the scores of the tests applied between the groups with dementia, MCI, DCS, and normal, the dementia group had worse scores in the MMSE, total ACE-R, and all sub-items, besides higher scores in the Pfeffer and CDR (Appendix A). A comparison of the test scores applied between the normal and cognitive decline groups revealed no difference (Appendix A). In comparing the test scores applied between the groups with and without cognitive decline (CD), the CD group had lower MMSE, total ACE-R, and ACE-R sub-item scores (Appendix A). A comparison of *APOE* genotyping and its haplotypes between dementia, MCI, DCS, and normal groups revealed no difference between groups (Appendix A). The comparison of genotyping and *APOE* alleles between groups with and without cognitive decline showed no differences (Appendix A).

## 4. Discussion

This study examined a group of outpatients experiencing post-COVID-19 neurological symptoms. Cognitive alterations were the primary concern, even in mild cases. Moreover, the ε4 allele was more frequent in the cognitive decline group. Furthermore, sleep disorders and anxiety symptoms were more common in the cognitive decline group. Our study found a higher, statistically significant frequency of the *APOE* ε4 allele in the cognitive decline group than in the normal group, and the *APOE* ε4 allele was an independent risk factor for CD. We speculate that this different result derives from the increase in participants and, mainly, that this new study grouped patients with MCI, SCD, and dementia under the term CD, allowing us to compare patients with cognitive complaints versus those without. To date, the studies that cited *APOE*’s participation in the COVID-19 manifestations have focused on clinical manifestations, and few studies are showing the role of *APOE* polymorphism in the genesis of post-COVID-19 cognitive manifestations [5,6]. Kuo et al., for instance, compared the *APOE* gene polymorphism with COVID-19 infection by logistic regression in a cohort of 622 participants in the United Kingdom and showed that patients with the ε4/ε4 genotype were more likely to be infected by COVID-19 (OR = 2.31, 95% CI: 1.65–3.24, *p* = 1.19 × 10^−6^) regardless of history of diabetes, cardiovascular disease, or dementia [6]. Furthermore, Espinosa-Salinas et al. used multiple comparison tests and investigated the association between the *APOE* gene polymorphism and the risk of COVID-19 infection, documenting a protective effect of the ε2 allele (OR: 0.207; CI: 0.0796, 0.538; *p* = 0.001) [5]. 

Our study also found a higher frequency of sleep complaints and anxiety symptoms in the CD group. Patients with sleep complaints may have a higher frequency of DCS and a higher frequency of anxiety symptoms, as previously reported by Jessen et al. [13]. Moreover, sleep disorders, such as insomnia or excessive sleepiness, may accompany Long-COVID-19 [17]. 

Several hypotheses regarding the genesis of cognitive symptoms after COVID-19 have been formulated, including ischemic brain changes, endothelial injury, and inflammatory reactions [18,19]. This last finding is relevant, as microglial inflammation is associated with Alzheimer’s [20]. Concerning Alzheimer’s, there is evidence that the ε4 allele of *APOE* stimulates brain amyloidogenesis via increased production more than the other isoforms of *APOE* and increases tau hyperphosphorylation under stress [21]. Furthermore, a vital link can be created between our findings and recent pathophysiology findings related to neurologic COVID-19 symptoms and neurodegenerative diseases [22,23]. Crunfli et al. showed that post-COVID-19 neurological manifestations can be related to astrocytopathy [22]. Moreover, animal models suggest that the *APOE* ε4 allele may be to blame for microglial activation in the early stages of Alzheimer’s [23]. Ramani et al. evaluated brain organoid neurons and revealed that exposure to SARS-CoV-2 induces stress, whose response leads to aberrant tau protein phosphorylation and apparent neuronal death [24]. Yet Segev et al. provided evidence that the ε4 allele promotes memory impairment mediated by the integrated stress response [25]. Zhang et al., using cell culture and animal models, evaluated the role of *APOE* in the interaction of the spike protein of SARS-CoV-2 with ACE and subsequent entry into infected cells [8]. These authors showed a possible protective role of *APOE* concerning viral entry into cells, with a worse performance by the ε4 allele compared to the ε3 allele, probably due to the more compact structure of the ε4 allele and, therefore, to its fewer spatial interference in preventing the interaction between virus and cell [8]. Furthermore, Chen et al. performed a meta-analysis to evaluate the interaction between APOE, the spike protein, and ACE. The authors showed that the APOE ε4 allele downregulates ACE2 protein expression in vitro and in vivo and consequently decreases the conversion of Angiotensin II to Angiotensin 1–7, which may introduce a potential mechanism by which *APOE* ε4 is associated with COVID-19 severity [26]. Lastly, Fernández-de-las-Peñas et al. found no association between the *APOE* ε4 allele and the number of COVID-19 symptoms, despite having only included hospitalized patients [27]. Through these findings, we can propose that the *APOE* ε4 allele may contribute to the genesis of cognitive impairment in patients with long-term COVID-19 since it protects less against COVID-19 infection and stimulates a pro-inflammatory response in patients with COVID-19, reducing endothelial repair and antioxidant activity in these patients and inducing greater microglial activation. Furthermore, the possible development of cognitive impairment in patients with Long-COVID-19 who carry the *APOE* ε4 allele raises concerns about the later development of neurodegenerative diseases (mainly Alzheimer’s), given the known role of such an allele as a risk factor for sporadic Alzheimer’s, corroborated by animal models that show its role in inducing cerebral amyloidogenesis.

Our study did not find a significant difference between the applied cognitive functionality and psychiatric assessment scales. Other authors have also reported this poor performance of brief cognitive screening batteries in post-COVID-19 cognitive assessment [28,29,30]. Kumar Khanna et al. evaluated 284 patients in India, six months after infection, using the MoCA and found no global decline with that battery. They concluded by emphasizing the importance of a detailed neuropsychological assessment [28]. In turn, Lynch et al. compared the performance of the MoCA with a neuropsychological assessment evaluating 60 post-COVID-19 patients, and the MoCA was 63.3% accurate in detecting some degree of reduced neuropsychological performance [30]. While subjective, the complaints reported by patients in our study involved symptoms concerning the cognitive domains of attention, executive functions, and memory [31]. Studies with more detailed cognitive assessments also found in this cognitive profile an impairment in these cognitive domains [32,33]. García-Sanchez et al. evaluated 63 patients with subjective cognitive complaints more than three months after COVID-19 infection with an extensive neuropsychological assessment, denoting that the most affected cognitive domains were attention, executive functions, and memory [33]. Delgado-Alonso et al. examined 50 patients through a detailed neuropsychological evaluation, with a mean age of 51 years (SD = 11.65) and similar to our study, evaluated more than 6 months after infection, identifying attention, executive functions, and memory [32]. In other studies, the most affected cognitive domain was memory [34,35]. This is important since the limbic structures, the epicenter of the cognitive domain of memory, can be affected by conditions associated with neuroinflammation [36]. Likewise, memory complaints in patients with more severe clinical conditions may be caused by the hippocampus being sensitive to low oxygen concentrations [37]. In this sense, Hosp et al. evaluated PET-FDG of the skull in patients in the acute phase of COVID-19 and showed limbic involvement associated with other brain structures [38]. In two different publications, using PET-FDG of the skull, Hugon et al. evaluated patients with mild COVID-19 and subsequent Long-COVID-19 with impaired memory, attention, and executive dysfunction, pointing to hypometabolism in the pons in three cases and the cingulate cortex in another two cases [39,40]. The pons and the anterior cingulate are structures whose injuries can cause executive dysfunction, thus being a possible anatomical substrate responsible for part of the cognitive symptomatology in patients with Long-COVID-19 [31].

Older adults were more susceptible to severe COVID-19 manifestations throughout the pandemic, which also puts this population at risk of cognitive decline after such more clinically severe conditions and also after hospitalization [41]. However, our study did not find any influence of age on the cognitive complaints identified, which may be due to a relatively young mean age in our sample and because most of our sample comprised patients with mild and outpatient conditions.

Our study found a trend towards an inverse relationship between cognitive impairment and anosmia, which disagrees with other studies. Cristillo et al. found a direct association between cognitive impairment and olfactory disorders in patients with COVID-19. However, in an older population, it likely signaled a marker of brain aging similar to that found in other studies [42]. Finally, our study found no associations between cognitive impairment and headache. Notwithstanding this, the association between headache and cognitive impairment can be found in patients after the acute phase of COVID-19 [43].

Furthermore, the origin of cognitive complaints may be due to psychiatric disorders [44]. Likewise, depressive symptoms are commonly associated with cognitive complaints, as in SCD [13]. In our study, individuals with cognitive decline did not have a higher frequency of depression. Ishmael et al. evaluated patients with mild COVID-19 and showed that 26.2% of patients persisted with depressive symptoms two months after infection [45]. Likewise, the impact of the disease on patients’ quality of life may contribute to depressive symptoms [46].

Our study has some significant limitations. First, there was no control group. Second, we only included patients with neurological symptoms who came to us after an announcement in social media and the media, which indicates a selection bias. Third, we have yet to have a previous cognitive assessment of patients. Furthermore, we have not had a previous cognitive assessment of the patients. Also, the results represent data for a single country. Moreover, the gold standard for classifying patients into MCI or SCD involves a detailed neuropsychological assessment rather than the method used in this study, which is only through cognitive screening tests and targeted anamnesis. It is also crucial to assess how cognitive symptoms will behave after treatment for depression in those patients with this diagnosis. Furthermore, the presence of anxiety symptoms was more frequent in patients with CD, but this complaint was not evaluated on any objective scale. It is also important to point out the additional limitation of not having differentiated the complaints between those reported voluntarily and those who were questioned by the researcher, since the voluntary reporting of complaints may denote a more significant impact on the patient’s life. Likewise, the fact that the ε4 allele correlates with memory impairment associated with the lack of a control group in our study prevents us from determining the direct causal role of Long-COVID-19 in the cognitive manifestations of those carrying this allele. Finally, there was no neuroimaging evaluation, hindering associations between complaints and radiological correlations.

One of our study’s main strengths is the assessment of patients after the acute phase of the disease. Moreover, our sample consisted of young patients with a high level of education, factors linked to greater cognitive reserve, mild forms of the disease, and after the acute/subacute phases of the disease, allowing us to show persistent symptoms even in this population [47]. Post-COVID-19 cognitive manifestations in patients with mild forms and high cognitive reserve suggest a significant and greater direct role of COVID-19 as a causal factor. Furthermore, grouping patients with dementia, MCI, and SCD under the term CD showed the high frequency of cognitive complaints in the same way that it valued the subjective complaints brought by patients, which motivated the search for care, which was important since the subjective cognitive complaints reported during the pandemic were not initially valued. However, such complaints were later objectively confirmed, primarily when evaluated by a detailed neuropsychological assessment [33]. Moreover, the analysis of the *APOE* polymorphism and possible associations with cognitive symptoms is unprecedented in the literature and strengthens our study.

Our study contributes valuable insights into patients experiencing cognitive issues following COVID-19. We observed that cognitive complaints are prevalent among COVID-19 patients, persisting even after the acute phase and in mild cases. Notably, the ε4 allele was more common in the group with cognitive decline. Long-term monitoring of these patients is crucial to ascertaining the persistence of this cognitive impairment over time. Additionally, conducting comprehensive neuropsychological assessments is essential for thoroughly characterizing subjects with subjective cognitive decline (SCD) or mild cognitive impairment (MCI) and for identifying the most affected cognitive domains. Lastly, it is imperative to explore neurodegenerative disease biomarkers in cerebrospinal fluid or plasma among those with cognitive impairment, linking COVID-19 to the initiation or progression of neurodegenerative disorders [48].

## Figures and Tables

**Figure 1 brainsci-13-01611-f001:**
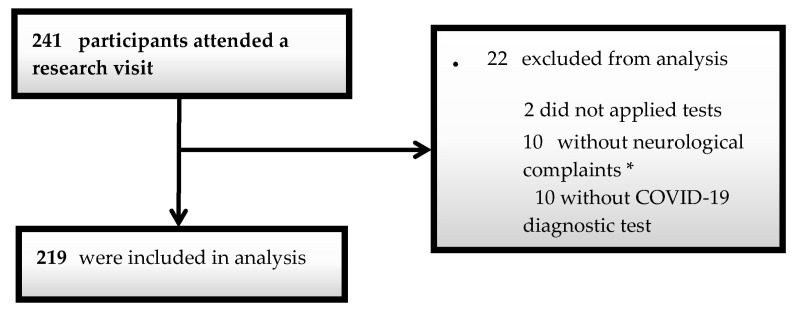
Flow diagram of participants. * may indicate headaches, anosmia, cognitive complaints, and others.

**Table 1 brainsci-13-01611-t001:** Sociodemographic and clinical characteristics of the sample, post-COVID-19 symptoms, and post-COVID-19 cognitive status.

Sociodemographic and Clinical Data	Total Group (*n* = 219)
** *Sociodemographic data* **	
**Gender**	
Female	142 (64.8)
Male	77 (35.2)
**Age (years)**	46.4 ± 14.5
**Age range**	
<50 years	130 (59.4)
50–65 years	67 (30.6)
>65 years	22 (10)
**Scholarity in Years**	
0–4 years	16 (7.3)
5–8 years	27 (12.3)
>8 years	176 (80.4)
**Hospitalization**	
No	164 (74.9)
Yes	55 (25.1)
** *Clinical data—post-COVID-19 symptoms* **	
**Anosmia**	
No	154 (70.3)
Yes	65 (29.7)
**Sleep disorders**	
No	155 (71)
Yes	64 (29)
**Depression**	
No	181 (82.6)
Yes	38 (17.4)
**Anxiety symptoms**	
No	162 (74)
Yes	57 (26)
**Headache**	
No	150 (68.5)
Yes	69 (31.5)
**Cognitive status**	
Dementia	11 (5)
MCI	25 (11.4)
SCD	107 (48.9)
Normal	76 (34.7)
**Cognitive decline (CD) x Normal**	
Normal	76 (34.7)
Cognitive decline (CD)	143 (65.3)
**Cognitive impairment (CI) x No Cognitive impairment**	
No cognitive impairment	183 (83.6)
Cognitive impairment (CI)	36 (16.4)

Continuous data are expressed as the mean ± standard deviation. Categorical data are expressed as absolute counts and percentages in parentheses. MCI: mild cognitive impairment; SCD: subjective cognitive decline. Source: Authors elaboration, 2023.

**Table 2 brainsci-13-01611-t002:** Sociodemographic, clinical characteristics, and post-COVID-19 symptoms according to cognitive status.

	Normal (*n* = 76)	Dementia (*n* = 11)	SCD (*n* = 107)	MCI (*n* = 25)	*p*-Value *
**Gender**					0.197
Female	45 (59.2)	5 (45.5)	76 (71)	16 (64)	
Male	31 (40,8)	6 (54,5)	31 (29)	9 (36)	
**Age (years)**	43.2 ± 14.2	69.8 ± 12.8	45.3 ± 12.7	50.3 ± 13.8	<0.001 #
**Age range**					<0.001
<50 years	54 (71.1)	1 (9.1)	63 (58.9)	12 (48)	
50–65 years	16 (21.1)	2 (18.2)	38 (35.5)	11 (44)	
>65 years	6 (7.9)	8 (72.7)	6 (5.6)	2 (8)	
**Scholarity in Years**					<0.001
0–4 years	6 (7.9)	5 (45.5)	5 (4.7)	0 (0)	
5–8 years	9 (11.8)	3 (27.3)	9 (8.4)	6 (24)	
>8 years	61 (80.3)	3 (27.3)	93 (86.9)	19 (76)	
**Hospitalization**					<0.001
No	60 (78.9)	4 (36.4)	88 (82.2)	12 (48)	
Yes	16 (21.1)	7 (63.6)	19 (17.8)	13 (52)	
** *Clinical data—post-COVID-19 symptoms* **					
**Anosmia**					0.018
No	49 (64.5)	11 (100)	72 (67.3)	22 (88)	
Yes	27 (35.5)	0 (0)	35 (32.7)	3 (12)	
**Sleep disorders**					0.001
No	63 (82.9)	11 (100)	67 (62.6)	14 (56)	
Yes	13 (17.1)	0 (0)	40 (37.4)	11 (44)	
**Depression**					0.089
No	66 (86.8)	11 (100)	84 (78.5)	20 (80)	
Yes	10 (13.2)	0 (0)	23 (21.5)	5 (20)	
**Anxiety symptoms**					0.052
No	63 (82.9)	10 (90.9)	72 (67.3)	17 (68)	
Yes	13 (17.1)	1 (9.1)	35 (32.7)	8 (32)	
**Headache**					0.065
No	55 (72.4)	11 (100)	69 (64.5)	15 (60)	
Yes	21 (27.6)	0 (0)	38 (35.5)	10 (40)	

Continuous data are expressed as the mean ± standard deviation. Categorical data are expressed as absolute counts and percentages in parentheses. *: as determined by a chi-square test for categorical data and an ANOVA test with Tukey’s post-test for age. # *p* < 0.05 between the “Dementia” group vs. other groups. SCD: subjective cognitive decline; MCI: mild cognitive impairment. Source: Authors elaboration, 2023.

**Table 3 brainsci-13-01611-t003:** Sociodemographic, clinical, and post-COVID-19 symptom characteristics in normal and cognitive decline (CD) groups.

	Normal	CD	*p*-Value
	(*n* = 76)	(*n* = 143)	
**Gender**			0.090
Female	45 (59.2)	97 (67.8)	
Male	31 (40.8)	46 (32.2)	
**Age (years)**	43 ± 14	48 ± 14	<0.001
**Age range**			0.035
<50 years	54 (71.1)	76 (53.1)	
50–65 years	16 (21.1)	51 (35.7)	
>65 years	6 (7.9)	16 (11.2)	
**Scholarity in Years**			0.962
0–4 years	6 (7.9)	10 (7)	
5–8 years	9 (11.8)	18 (12.6)	
>8 years	61 (80.3)	115 (80.4)	
**Hospitalization**			0.312
No	60 (78.9)	104 (72.7)	
Yes	16 (21.1)	39 (27.3)	
** *Clinical data—post-COVID-19 symptoms* **			
**Anosmia**			0.167
No	49 (64.5)	105 (73.4)	
Yes	27 (35.5)	38 (26.6)	
**Sleep disorders**			0.004
No	63 (82.9)	92 (64.3)	
Yes	13 (17.1)	51 (35.7)	
**Depression**			0.232
No	66 (86.8)	115 (80.4)	
Yes	10 (13.2)	28 (19.6)	
**Anxiety symptoms**			0.028
No	63 (82.9)	99 (69.2)	
Yes	13 (17.1)	44 (30.8)	
**Headache**			0.368
No	55 (72.4)	95 (66.4)	
Yes	21 (27.6)	48 (33.6)	

Continuous data are expressed as the mean ± standard deviation. Categorical data are expressed as absolute counts and percentages in parentheses. As determined by a chi-square test for categorical data and a Student’s *t*-test for age. CD: cognitive decline. Source: Authors elaboration, 2023.

**Table 4 brainsci-13-01611-t004:** APOE genotypes and alleles in the total group.

	(*n* = 185) * N (%)
** *APOE* **	
E2/E2	1 (0.55)
E2/E3	14 (7.7)
E2/E4	1 (0.55)
E3/E3	122 (65.9)
E3/E4	43 (23.2)
E4/E4	4 (2.2)
**Alleles**	
**E2**	
No	169 (91.3)
Yes	16 (8.6)
**E3**	
No	6 (3.2)
Yes	179 (96.7)
**E4**	
No	137 (74.05)
Yes	48 (25.9)

Categorical data are expressed as absolute counts and percentages in parentheses. APOE: Apolipoprotein E gene. *: patients who had blood drawn for APOE polymorphism analysis. Source: Authors elaboration, 2023.

**Table 5 brainsci-13-01611-t005:** Comparison of APOE genotypes and alleles between normal and cognitively impaired groups.

	Normal (*n* = 61)	CD (*n* = 124)	*p*-Value
** *APOE* **			0.391
E2/E2	0 (0)	1 (0.8)	
E2/E3	5 (8.2)	9 (7.3)	
E2/E4	0 (0)	1 (0.8)	
E3/E3	46 (75.4)	76 (61.3)	
E3/E4	9 (14.8)	34 (27.4)	
E4/E4	1 (1.6)	3 (2.4)	
**Alleles**			
**E2**			0.878
No	56 (91.8)	113 (91.1)	
Yes	5 (8.2)	11 (8.9)	
**E3**			0.665
No	1 (1.6)	5 (4)	
Yes	60 (98.4)	119 (96)	
**E4**			0,038
No	51 (83.6)	86 (69.4)	
Yes	10 (16.4)	38 (30.6)	

Categorical data are expressed as absolute counts and percentages in parentheses. The chi-square or Fisher’s exact test was used to determine statistical significance. CD: cognitive decline; APOE: apolipoprotein E. Source: Authors elaboration, 2023.

**Table 6 brainsci-13-01611-t006:** Multivariate logistic regression evaluating the independent association of the presence of allele E4 with DC adjusted for other confounders.

	DC			
	Initial Model		Final Model	
	Odds Ratio (95% CI)	*p*-Value	Odds Ratio (95% CI)	*p*-Value
**Sex (male)**	0.571 (0.282; 1.158)	0.120		
**Age**				
<50 years	-	0.047	-	0.052
50–65 years	2.284 (1.03; 5.061)	0.042	2.102 (0.988; 4.475)	0.054
>65 years	4.062 (0.911; 18.112)	0.066	3.077 (0.923; 10.254)	0.067
**Anxiety symptoms (yes)**	3.811 (1.576; 9.213)	0.003	3.758 (1.58; 8.938)	0.003
**Anosmia (yes)**	0.597 (0.28; 1.273)	0.182		
**Education**				
0–4 years	-	0.625		
5–8 years	1.109 (0.189; 6.514)	0.909		
9 years or more	1.776 (0.331; 9.542)	0.503		
**Sleep disorder (yes)**	2.029 (0.892; 4.618)	0.092	1.931 (0.877; 4.255)	0.102
**Presence of allele E4**	2.015 (0.839; 4.838)	0.117	2.336 (1.035; 5.272)	0.041
**Presence of allele E3**	0.501 (0.045; 5.603)	0.575		
**Presence of allele E2**	1.138 (0.329; 3.94)	0.839		

The stepwise backward method was used to reach the final model.

**Table 7 brainsci-13-01611-t007:** Comparison of test scores applied to the total group according to the age group of the patients.

	Age			
	<50 Years	50–65 Years	>65 Years	*p*-Value
**PFEFFER median (Min–Max)**	0 (0–30)	0 (0–20)	0 (0–30)	<0.001 ^A^
**Beck’s Depresion Inventory (IQR)**	0 (0–10)	2 (0–12)	0 (0–17)	0.801
**GDS (IQR)**	0 (0–0)	4 (0–6)	3 (1–5)	0.455
**CDR, median (Min–Max)**	0 (0–0)	0 (0–1)	0 (0–3)	<0.001 ^A^
**PRMQ, mean ± SD**	7 ± 3	6 ± 2	10 ± 8	<0.001 ^A^
**MMSE, mean ± SD**	27.9 ± 4.1	26.3 ± 6.4	20.3 ± 10.1	<0.001 ^A^
**ACE-R, mean ± SD**	84.9 ± 14.2	79.7 ± 20.9	54.4 ± 31.1	<0.001 ^A^
**Orientation/Attention, mean ± SD**	16.8 ± 2.7	16.1 ± 3.9	12.2 ± 6.5	<0.001 ^A^
**Memory, mean ± SD**	19.6 ± 5.1	19 ± 5.7	12.3 ± 8.5	<0.001 ^A^
**Verbal fluency, mean ± SD**	10.2 ± 2.9	9.5 ± 3.3	5.4 ± 3.9	<0.001 ^A^
**Language, mean ± SD**	24.1 ± 3.8	22.6 ± 6.3	16 ± 9.4	<0.001 ^A^
**Visuospatial abilities, mean ± SD**	14.2 ± 2.8	12.9 ± 4.4	8.6 ± 5	<0.001 ^B^

IQR: interquartile range. SD: standard deviation. As determined by an ANOVA test with Tukey’s post-test for means and the Kruskal-Wallis test with multiple comparisons for medians. A: *p* < 0.05: >65 years old vs. other groups; B: *p* < 0.05: between all groups. GDS: Geriatric Depression Scale; CDR: Clinical Dementia Rating; PRMQ: Prospective and Retrospective Memory Questionnaire’s; MMSE: Mini Mental State Examination; ACE-R: Addenbrooke´s Cognitive Examination-Revised.

## Data Availability

Data is available upon request.

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
