# Peer review of "Post-COVID-19 Cognitive Decline and Apoe Polymorphism: Towards a Possible Link?"

_brainsci, 2023, doi:10.3390/brainsci13121611_

Round 1

Reviewer 1 Report

Comments and Suggestions for Authors

This manuscript entitled “POST-COVID COGNITIVE DECLINE AND APOE POLIMORPHISM : TOWARDS A POSSIBLE LINK ?” has been reviewed sincerely.

The topic of this study is a clinically significant and interesting. Although I have no serious comments in this submitted paper, there were several concerns as below.

1.  In the Title, “POLIMORPHISM” may bePOLYMORPHISM”.

2. There are many different races in Brazil. Differences of their race may influence the results. The race of patients should be provided at first.

3. Authors should describe about limitation of this study, i.e. “the study showed the results of single country”.

4. Please discuss about relationship between severity of COVID-19 infection and ApoE polymorphism.  

5. Did the authors confirm the inter-examiner error? Authors need to indicate if the examiners were standardized and how many examiners performed the clinical exams.

Author Response

Date: 13/11/2023

Title: “POST-COVID COGNITIVE DECLINE AND APOE POLYMORPHISM : TOWARDS A POSSIBLE LINK ?”

Dear Editors,

We are very pleased to review and resubmit our manuscript and comment on the suggestions and critics made by the reviewers. We sincerely and technically hope to clarify pending questions raised by the reviewers.

Reviewer comments are raised by yellow and changes performed are highlighted in red in the main manuscript. 

-------------------------------------------------------------------------------------------------------

Reviewer #1

  1. In the Title, “POLIMORPHISM” may be “POLYMORPHISM”.

Response: We are sorry for this typo. The manuscript was reviewed.

  1. There are many different races in Brazil. Differences of their race may influence the results. The race of patients should be provided at first.

Response: We did not determine the races of patients. We now make this clear in the manuscript at Methods section.

  1. Authors should describe about limitation of this study, i.e. “the study showed the results of single country”.

Response: We agree with the reviewer. We now added this sentence below:

Also, the results represent data in a single country.

  1. Please discuss about relationship between severity of COVID-19 infection and ApoE polymorphism.  

Response: We agree with the reviewer. We added these sentences below at Introduction:

Kuo et al. linked more severe COVID-19 in subjects with  E4 allele of the APOE gene. The authors of this study hypothesized if  this finding might  be  related to the high level of expressed of APOE genes together with angiotensin-converting enzyme 2 (ACE-2) in the alveolar cells of the lungs’.7 Another group of researchers studied 249 volunteers with an average age of 49 years and demonstrated the protective role of the E2 allele against more severe clinical conditions of COVID-19.6 Similarly, Zhang et al. evaluated 142 patients with COVID-19 and found that those with APOE E4 had elevated inflammatory factors.9 Wang et al. demonstrated that SARS-CoV-2 infection preferentially activates astrocytes, promotes greater reduction in neurite length and causes a greater cytopathogenic effect on these cells, in addition to inducing greater death of astrocytes carrying the E4 allele compared to those carrying E3.10 In another study, Zorkina et al. did not find any influence of the baseline serological status for COVID-19 and the APOE gene polymorphism on cognitive rehabilitation in a sample of individuals over 65 years old measured through changes in Mini Mental State Examination (MMSE) scores.11 This association is significant, as the same allele confers a higher risk of sporadic Alzheimer's disease (AD).12

And these in Discussion section:

Kuo et al., for instance , compared the APOE gene polymorphism with COVID-19 infection by logistic regression in a cohort of 622 participants in the United Kingdom and demonstrated that patients with the ε4/ε4 genotype were more likely to be infected by COVID-19 (OR=2.31, 95% CI: 1.65-3.24, p=1.19 × 10-6) regardless of history of diabetes, cardiovascular disease or dementia.7 Furthermore, Espinosa-Salinas et al. used multiple comparison tests and investigated the association between the APOE gene polymorphism and the risk of COVID-19 infection, documenting a protective effect of the ε2 allele (OR: 0.207; CI: 0.0796, 0.538; p= 0.001).6 

  1. Did the authors confirm the inter-examiner error? Authors need to indicate if the examiners were standardized and how many examiners performed the clinical exams.

Response: We now make clear in the Methods section that only 2 trained neurologists administered the assessments. We did not evaluted the inter-examiner error.

Only two independent neurologists (JWLTJ and DNO) performed patient clinical evaluations. We did not evaluated the inter-examiner error.

Reviewer 2 Report

Comments and Suggestions for Authors

The main idea of the manuscript by Tavares-Júnior is to establish a link between COVID-19-related cognitive impairment and APOE gene polymorphism in a Brazil patients.

Many studies are aimed at finding associations with the diagnosis of mild congnitive impairment (MCI) or Alzheimer disease (AD), with a confirmed APOE genotype. However, the degree of cognitive impairment progression in individuals with different APOE genotypes, as well as studies of impairment in specific cognitive domains, are of greater practical interest.

In fact, it was described that the APOE gene polymorphism can affect susceptibility to SARS-CoV-2 infection, COVID-19 severity and mortality (Espinosa-Salinas et al, 2022, Sci. Rep; Kuo et al, 2020, J. Gerontol. A. Biol. Sci. Med. Sci.; Manzo et al, 2021, Med. Hypotheses). The APOE ε4 allele was a risk factor for severe COVID-19 and post-COVID mental fatigue (Kurki et al, 2021, Acta Neuropathol Commun; Kuo et al, 2020, J. Gerontol. A. Biol. Sci. Med. Sci.), whereas protective effect of ε2 allele against SARS-CoV-2 infection was shown (Espinosa-Salinas et al, 2022, Sci. Rep.). In the brain autopsy material of COVID-19 patients, perivascular microhemorrhages were found to be more common in APOE ε4 carriers, suggesting that some of these effects may be mediated by increased cerebrovascular damage (Kurki et al, 2021, Acta Neuropathol Commun). A recent study by Zorkina et al showed a positive effect of cognitive training that depended on the APOE genotype in patients with MCI (Zorkina et al, 2022, Diagnostics).

While the authors' research is important and necessary, I am surprised that the authors state that no one has found an association between COVID-19 cognitive decline and APOE genotype (see, for example lines 18-19 of the abstract). This is not entirely true.

Overall, the paper presents important experimental findings. However, it requires extensive revision to enhance its structure, presentation rigor, and visualization elements. Additionally, using consistent terminology, precise language, and adhering to measurement standards would benefit the clarity of the manuscript.

General comments to authors: please, italicize the APOE throughout the manuscript, since you are mentioned the APOE gene polymorphism, not the ApoE protein. Please define the abbreviation once and avoid deciphering it repeatedly throughout the text of the manuscript.

Abstract

According to the journal requirements rhe abstract should be a total of about 200 words maximum. Abstract should be shortened.

The abstract should be a single paragraph and should follow the style of structured abstracts, but without headings.

Line 17. Please replace APOE4 with APOE-ε4

Please indicate the methods or scales utilized to assess neurological symptoms in COVID-19 patients because the following statement is unclear ‘However, this complaint was objectively verified through screening tests in only 36 patients (16.4%)’ (lines 28-29).

Lines 31 and 34. There is no need to emphasize ε4 with bold text.

Introduction

The "Introduction" section consists of 13 lines. In my opinion, it is too laconic.

The authors should conduct a comprehensive review of the effects of coronavirus infection on cognitive function. They can then discuss the impact of APOE on the development of dementia and its contribution to cognitive decline before transitioning to the statement of the problem.

For example, a study conducted by in South Korea evaluated the number of newly diagnosed cases of dementia and the worsening of comorbid psychiatric symptoms in subjects with dementia who tested positive for COVID-19 (Kim et al, 2022. Am J Alzheimers Dis Other Demen.). A meta-analysis of Soysal et al. suggests that neuropsychiatric symptomatology is present and worsens with COVID-19 isolation and pandemic in patients with dementia (Soysal et al, 2022, Psychogeriatrics), etc.

Line 40. Please remove an excessive dot.

Lines 47-48. ‘An article published with the preliminary results of our study did not reveal an association between cognitive impairment and APOE’. This statement is supported by study by Petersen et al, 1999, Archives of neurology. Hasn't there been any new research on this topic since 1999? For example, Caselli et al, 2004, Neurology; Whitehair et al, 2010, Alzheimers Dement;  Alegret et al, 2014, . J Alzheimers Dis; Qian et al, 2021, Neurology, etc. I suggest that the authors thoroughly address this issue before proceeding with the experimental aspect of it.

Materials and Methods

This section must be re-written. Please, provide separate sub-sections regarding:

- subjects (patients cohorts); How was the control group defined by the authors?

- APOE genotyping. How the authors determined the ε-alleles of APOE ? Primer sequences, or references to articles, etc.;

- methods to access the cognitive status of patients;

- statistical processing – very important for such studies.

Not really sure why the sentence on lines 79-81 is highlighted in red.

The current sub-section 2.2. ‘Ethical Aspects’ should be transferred to the corresponding section ‘Institutional Review Board Statement’ after the main part of manuscript.

Results

It is recommended to divide this section into sub-sections with appropriate headings for better clarity and organization. The purpose of providing the same figures and tables in the Supplementary as in the manuscript is unclear.

What is about a control group?

However, the Supplementary file contains valuable information that could be incorporated into the main text. Tables 7-13 of the Supplementary can be referred to as Table Sn (n = 1-7).

Line 158. The authors now use the term Cognitive impairment (CI) instead of the previously used Cognitive decline (CD).

I suggest including a table in the Results chapter that compares patient age (possibly by cohort), ACE-R, MMSE, CDR scores, and APOE genotypes. This would provide clarity and efficiency, as opposed to the currently lengthy Tables 2-5.

Discussion

The section looks substantial, but some of the information from it should be moved to Introduction (lines 192-202) and Materials and Methods (lines 189-190) sections.

I am surprised that no age-related effect on cognitive change was detected in COVID patients by the authors. Older adults are at high risk of developing severe forms of COVID-19 because of factors associated with aging and a higher prevalence of comorbid medical conditions, and therefore they are more vulnerable to possible long-term neuropsychiatric and cognitive impairment. It is possible that this was due to the selected cohort, but an explanation in the Discussion section would be beneficial.

Another limitation of the article, which may also be discussed, is that the authors were unable to conduct prior cognitive assessments of patients and control groups.

Line 289 - ??? after ref#38.

Lines 329-331. Study of biomarkers in cerebrospinal fluid or plasma among those with cognitive impairment is important. I agree with it. An example of such study was recently published by Abramova et al, 2023, Int. J. Mol Sci.

Comments on the Quality of English Language

Moderate editing of English language required

Author Response

Date: 13/11/2023

Title: “POST-COVID COGNITIVE DECLINE AND APOE POLYMORPHISM : TOWARDS A POSSIBLE LINK ?”

Dear Editors,

We are very pleased to review and resubmit our manuscript and comment on the suggestions and critics made by the reviewers. We sincerely and technically hope to clarify pending questions raised by the reviewers.

Reviewer comments are raised by yellow and changes performed are highlighted in red in the main manuscript. 

-------------------------------------------------------------------------------------------------------

Reviewer #2

While the authors' research is important and necessary, I am surprised that the authors state that no one has found an association between COVID-19 cognitive decline and APOE genotype (see, for example lines 18-19 of the abstract). This is not entirely true.

Response: We agree with the reviewer. We corrected it. The following sentence was removed from Abstract and Introduction:

As far as we know, there are no publications that have identified significant associations between COVID-19 and APOE polymorphism.

We additionally  described studies on the relationship between APOE and Covid-19, including post-Covid cognitive decline, as below:

In another study, Zorkina et al. did not find any influence of the baseline serological status for COVID-19 and the APOE gene polymorphism on cognitive rehabilitation in a sample of individuals over 65 years old measured through changes in Mini Mental State Examination (MMSE) scores.11

General comments to authors: please, italicize the APOE throughout the manuscript, since you are mentioned the APOE gene polymorphism, not the ApoE protein. Please define the abbreviation once and avoid deciphering it repeatedly throughout the text of the manuscript.

Response: We are sorry for these issues. The manuscript was reviewed.

Abstract

According to the journal requirements the abstract should be a total of about 200 words maximum. Abstract should be shortened. The abstract should be a single paragraph and should follow the style of structured abstracts, but without headings.

Response: We appreciated this comment and reformulated the abstract.

Line 17. Please replace APOE4 with APOE-ε4

Response: We are sorry for these issues. The manuscript was reviewed.

Please indicate the methods or scales utilized to assess neurological symptoms in COVID-19 patients because the following statement is unclear ‘However, this complaint was objectively verified through screening tests in only 36 patients (16.4%)’ (lines 28-29).

Response: We thank the reviewer for this comment. We make it clear now. We explain the applied scales as below:

(Addenbrooke Cognitive Examination-Revised and Mini Mental State Examination)

Lines 31 and 34. There is no need to emphasize ε4 with bold text.

Response: We are sorry for these issues. The manuscript was reviewed.

Introduction

The "Introduction" section consists of 13 lines. In my opinion, it is too laconic. The authors should conduct a comprehensive review of the effects of coronavirus infection on cognitive function. They can then discuss the impact of APOE on the development of dementia and its contribution to cognitive decline before transitioning to the statement of the problem.

Response: We totally agree with the reviewer. We now corrected it. We added these sentences below:

Subsequently, cognitive manifestations after the acute and subacute phases of the disease began to be reported even in patients with mild or asymptomatic forms of the disease. Such manifestations can normally occur with other symptoms, such as fatigue and sleep disturbances, in a condition that has been called Long-Covid.4 Brutto et al. in this sense evaluated outpatients with mild cases of the disease 6 months after infection and, using the MoCA, demonstrated a decline in 21% of patients compared to data from the same patients before the pandemic.3

Kuo et al. linked more severe COVID-19 in subjects with ε4 allele of the APOE gene. The authors of this study hypothesized if  this finding might  be  related to the high level of expressed of APOE genes together with angiotensin-converting enzyme 2 (ACE-2) in the alveolar cells of the lungs’.6 Another group of researchers studied 249 volunteers with an average age of 49 years and demonstrated the protective role of the E2 allele against more severe clinical conditions of COVID-19.5 Similarly, Zhang et al. evaluated 142 patients with COVID-19 and found that those with APOE E4 had elevated inflammatory factors.8 In another study, Zorkina et al. did not find any influence of the baseline serological status for COVID-19 and the APOE gene polymorphism on cognitive rehabilitation in a sample of individuals over 65 years old measured through changes in Mini Mental State Examination (MMSE) scores.9 This association is significant, as the same allele confers a higher risk of sporadic Alzheimer's disease (AD).10

Thus, the possible post-Covid cognitive decline, in addition to the relationship between the APOE polymorphism and post-Covid severe and cognitive conditions, raises concerns regarding the subsequent development of neurodegenerative diseases, such as Alzheimer's disease.12

Line 40. Please remove an excessive dot.

Response: We are sorry for these issues. The manuscript was reviewed.

Lines 47-48. ‘An article published with the preliminary results of our study did not reveal an association between cognitive impairment and APOE’. This statement is supported by study by Petersen et al, 1999, Archives of neurology. Hasn't there been any new research on this topic since 1999? For example, Caselli et al, 2004, Neurology; Whitehair et al, 2010, Alzheimers Dement;  Alegret et al, 2014, . J Alzheimers Dis; Qian et al, 2021, Neurology, etc. I suggest that the authors thoroughly address this issue before proceeding with the experimental aspect of it.

Response: We are sorry for these issues. The manuscript was reviewed, including references.

Materials and Methods

This section must be re-written. Please, provide separate sub-sections regarding:

Response: We agree with the reviewer. We now added these sub-sections.

- subjects (patients cohorts); How was the control group defined by the authors?

Response: We did not have a control group. Unfortunately, the country was going through a serious health crisis during the pandemic, and patients without Covid-19 were afraid to participate in the research in a hospital environment. We now make this clear in the manuscript.

- APOE genotyping. How the authors determined the ε-alleles of APOE ? Primer sequences, or references to articles, etc.;

Response: We now added these sentences below:

ApoE genotypes were determined by real-time Polymerase Chain Reaction (qPCR) using the TaqMan® allelic discrimination system (TaqMan® SNP Genotyping Assay, ThermoFisher®).17 For this, we used probes according to the sequences provided by the manufacturer: C___3084793_20 (rs429358) and C____904973_10 (rs7412), observing the information contained in the catalog number: 4351379 and similar protocols described in the literature for performing the technique.”

- methods to access the cognitive status of patients; - statistical processing – very important for such studies.

Response: We now added these sub-sections.

Not really sure why the sentence on lines 79-81 is highlighted in red.

Response: We are sorry for thuis typo. The manuscript was reviewed.

The current sub-section 2.2. ‘Ethical Aspects’ should be transferred to the corresponding section ‘Institutional Review Board Statement’ after the main part of manuscript.

Response: We agree with the reviewer. We now corrected it.

Results

It is recommended to divide this section into sub-sections with appropriate headings for better clarity and organization. The purpose of providing the same figures and tables in the Supplementary as in the manuscript is unclear. However, the Supplementary file contains valuable information that could be incorporated into the main text. Tables 7-13 of the Supplementary can be referred to as Table Sn (n = 1-7).

Response: We agree with the reviewer. We now corrected it.

What is about a control group?

Response: We did not have a control group. Unfortunately the country was going through a serious health crisis during the pandemic, and patients without Covid-19 were afraid to participate in the research in a hospital environment. We now make this clear in the manuscript.

Line 158. The authors now use the term Cognitive impairment (CI) instead of the previously used Cognitive decline (CD).

Response: We are sorry for these issues. The manuscript was reviewed and these mistakes were corrected.

I suggest including a table in the Results chapter that compares patient age (possibly by cohort), ACE-R, MMSE, CDR scores, and APOE genotypes. This would provide clarity and efficiency, as opposed to the currently lengthy Tables 2-5.

Response: We appreciated this suggestion. We added a new table 7.

Discussion

The section looks substantial, but some of the information from it should be moved to Introduction (lines 192-202) and Materials and Methods (lines 189-190) sections.

Response: We appreciated this suggestion. We moved these sentences to the indicated sections.

I am surprised that no age-related effect on cognitive change was detected in COVID patients by the authors. Older adults are at high risk of developing severe forms of COVID-19 because of factors associated with aging and a higher prevalence of comorbid medical conditions, and therefore they are more vulnerable to possible long-term neuropsychiatric and cognitive impairment. It is possible that this was due to the selected cohort, but an explanation in the Discussion section would be beneficial.

Response: We agree with the reviewer. We now added this sentence below:

Throughout the pandemic, the elderly population proved to be more susceptible to serious manifestations of Covid-19. This fact also puts this population at risk of cognitive decline after such more serious clinical conditions, as well as after hospitalization. Our study, however, did not find any influence of age on the cognitive complaints found. We speculate that this may be due to a relatively young average age in our sample, in addition to the fact that the majority of our sample was composed of patients with mild and outpatient conditions.

Another limitation of the article, which may also be discussed, is that the authors were unable to conduct prior cognitive assessments of patients and control groups.

Response: We agree with the reviewer. We now added this sentence below:

Furthermore, we did not have a previous cognitive assessment of the patients.

Line 289 - ??? after ref#38.

Response: We are sorry for these issues. The manuscript was reviewed and these mistakes were corrected.

Lines 329-331. Study of biomarkers in cerebrospinal fluid or plasma among those with cognitive impairment is important. I agree with it. An example of such study was recently published by Abramova et al, 2023, Int. J. Mol Sci.

Response: We appreciated this suggestion. We added this reference below:

Abramova, O. et al. Alteration of Blood Immune Biomarkers in MCI Patients with Different APOE Genotypes after Cognitive Training: A 1 Year Follow-Up Cohort Study. Int J Mol Sci 24, (2023).

Finally, the English was revised and the translator's proofreading certificate is attached.

Round 2

Reviewer 1 Report

Comments and Suggestions for Authors

None

Author Response

Thanks for the suggestions.

Reviewer 2 Report

Comments and Suggestions for Authors

The authors have done a lot of work to improve the manuscript. They have responded to all the reviewer's comments. At the moment I have no complaints about the essence of the work. However, the reviewed version of the manuscript brainsci-2702571-peer-review-v4 is missing Tables and Figure 1, as well as references to mention Tables S1-S7. This looks weird. In addition, the list of references is not made according to MDPI journal rules. Some of the references do not contain the year of publication. The authors need to make the 'References' section in accordance with the journal rules.

Author Response

Date: 17/11/2023

Title: “POST-COVID COGNITIVE DECLINE AND APOE POLYMORPHISM : TOWARDS A POSSIBLE LINK ?”

Dear Editors,

We are very pleased to review and resubmit our manuscript and comment on the suggestions and critics made by the reviewers. We sincerely and technically hope to clarify pending questions raised by the reviewers.

Reviewer comments are raised by yellow and changes performed are highlighted in red in the main manuscript. 

The authors have done a lot of work to improve the manuscript. They have responded to all the reviewer's comments. At the moment I have no complaints about the essence of the work. However, the reviewed version of the manuscript brainsci-2702571-peer-review-v4 is missing Tables and Figure 1, as well as references to mention Tables S1-S7. This looks weird. In addition, the list of references is not made according to MDPI journal rules. Some of the references do not contain the year of publication. The authors need to make the 'References' section in accordance with the journal rules.

Response: We are sorry for these issues. The tables and figure were now inserted in the manuscript. The supplemental tables are now mentioned. The manuscript was reviewed, including references.
